# *QuickPic AAC*: An AI-Based Application to Enable Just-in-Time Generation of Topic-Specific Displays for Persons Who Are Minimally Speaking

**DOI:** 10.3390/ijerph21091150

**Published:** 2024-08-29

**Authors:** Christina Yu, Ralf W. Schlosser, Maurício Fontana de Vargas, Leigh Anne White, Rajinder Koul, Howard C. Shane

**Affiliations:** 1Boston Children’s Hospital, Waltham, MA 02453, USA or r.schlosser@northeastern.edu (R.W.S.); leighanne.white@childrens.harvard.edu (L.A.W.); howard.shane@childrens.harvard.edu (H.C.S.); 2Massachusetts General Hospital Institute of Health Professionals, Boston, MA 02129, USA; 3Department of Communication Sciences and Disorders, Northeastern University, Boston, MA 02115, USA; 4School of Information Studies, McGill University, Montreal, QC H3A 0G4, Canada; mauricio.fontanadevargas@mail.mcgill.ca; 5Department of Speech, Language, and Hearing Sciences, University of Texas at Austin, Austin, TX 78712, USA; rajinder.koul@austin.utexas.edu

**Keywords:** artificial intelligence, augmentative and alternative communication, AI, AAC, application, just-in-time, speech–language pathology

## Abstract

As artificial intelligence (AI) makes significant headway in various arenas, the field of speech–language pathology is at the precipice of experiencing a transformative shift towards automation. This study introduces *QuickPic AAC*, an AI-driven application designed to generate topic-specific displays from photographs in a “just-in-time” manner. Using *QuickPic AAC*, this study aimed to (a) determine which of two AI algorithms (NLG-AAC and GPT-3.5) results in greater specificity of vocabulary (i.e., percentage of vocabulary kept/deleted by clinician relative to vocabulary generated by *QuickPic AAC*; percentage of vocabulary modified); and to (b) evaluate perceived usability of *QuickPic AAC* among practicing speech–language pathologists. Results revealed that the GPT-3.5 algorithm consistently resulted in greater specificity of vocabulary and that speech–language pathologists expressed high user satisfaction for the *QuickPic AAC* application. These results support continued study of the implementation of *QuickPic AAC* in clinical practice and demonstrate the possibility of utilizing topic-specific displays as just-in-time supports.

## 1. Introduction

With the advent of mobile technology, the use of applications (“Apps”) in augmentative and alternative communication (AAC) has become integral to the standard of care for persons who are minimally speaking [1] (individuals who are minimally speaking may include persons with developmental disabilities (e.g., autism, intellectual disabilities, etc.), acquired disorders (e.g., aphasia, traumatic brain injury), progressive disorders (e.g., muscular dystrophy), and temporary conditions (e.g., recovering from surgery in the intensive care unit) [2]). Many apps provide a range of tools that serve as a communication platform as well as a medium to provide language support [3]. *QuickPic AAC* is a new and innovative app that seamlessly blends artificial intelligence (AI) and visual supports to empower minimally speaking individuals who require support generating utterances by selecting graphic representations or text from a display.

*QuickPic AAC* harnesses the power of AI to interpret visual scenes from a photograph, allowing it to identify characters and their actions. The source of the picture scenes can be a photo library, a fresh photo snapshot, or an internet search. *QuickPic AAC* then transforms the visual input into a mixed display which is a combination of the visual scene (photo) and the elements of vocabulary thematically related to the scene arranged in a grid display [4]. The grid display is arranged in the form of a modified Fitzgerald Key that parses and color-codes the grammatical parts of a sentence [5,6]. *QuickPic AAC* has the following categories from left to right: pronouns, verbs, prepositions, adjectives, and objects. In other words, after analyzing the photo, *QuickPic AAC* constructs a grid that strategically places symbols representing the subjects and their activities in the scene. Notably, the app uses facial recognition to recognize individuals and retains this knowledge to accurately identify individuals in future mixed displays. *QuickPic AAC* also allows instructors to edit and customize the symbols in the grid, ensuring the most accurate representation of the scene. This collaborative and customizable aspect guarantees that the app’s generated vocabulary not only aligns with the visual content, but is also personalized and meaningful to its user, enabling learners to better grasp language concepts.

The thematic or topic-specific vocabulary that is arranged in grammatical categories (i.e., based on the Fitzgerald Key) is known as a topic-specific display (TSD). TSDs are a type of aided approach that enable users to communicate appropriately with phrase production in the context of a particular activity [7,8] through the arrangement of linguistic elements or categories on a single page for constructing a sentence. Several studies have demonstrated the importance of considering symbol background color and/or grammatical category when organizing a display [9,10]. More specifically, these studies concluded that “clustering” symbols based on shared background color and/or grammatical category (e.g., subject, verb, noun, adjective) increases the speed at which early communicators can locate a desired symbol. The layout of the mixed displays created in *QuickPic AAC* reflect these findings, as shown by the organization of the grid display shown in Figure 1.

Thistle and Wilkinson [11] sought out to better understand how speech–language pathologists make decisions regarding display type and layout characteristics of AAC systems for early communicators. While 83% of participants indicated they use visual scene displays (VSDs) less than 25% of the time, 60% of participants indicated they would use a VSD for an early communicator rather than a grid-based display. This finding may suggest that while many speech–language pathologists see the value in utilizing a scene-based approach, the time it takes to create them may interfere with the frequency at which these valuable tools are used. Thistle and Wilkinson’s study [11] also captured how much variance there is among speech–language pathologists in their desired characteristics for AAC displays created for early communicators. For example, several participants of their study indicated preferences regarding the vantage point of photographs used, the background of photographs, and the size of the display. *QuickPic AAC* aims to make creating displays incorporating real photographs much more efficient, and therefore more likely to be used in clinical practice. If successful, this may also afford speech–language pathologists endless opportunities for customization to meet the specific needs of the individual.

Traditionally, developing meaningful and functional TSDs has required significant advanced planning and programming, and, therefore, time, from mentors working with individuals who are minimally speaking. For example, imagine a teacher is planning to introduce a new science lesson on a particular forest biotope in the weeks ahead. In addition to planning the lesson in general terms, this teacher would need to gather all the vocabulary needed for the student who is minimally speaking, and then organize it in an intuitive way, so that the student can be an effective participant in that lesson. 

Because the *QuickPic AAC* app enables automatic generation and organization of vocabulary from a single photograph, one could upload a photo of a forest biotope and the app would automatically generate and organize the vocabulary in the form of graphic symbols (e.g., Picture Communication Symbols). If functional, the mentor would save considerable time and elevate TSDs into the realm of just-in-time supports (JITs) [12,13]. This is something that was previously unthinkable given the advanced planning and time-consuming preparations. 

There is a significant gap in research related to AAC applications and their use in clinical practice. While there is a dearth of evidence supporting the effectiveness of speech-generating devices (SGDs) [14,15,16] as a general category of effective communication tools, there is limited research on the specific features, functions, and algorithms used in AAC applications. In addition, there is significant growing interest and discussion of the use of AI in the field of speech–language pathology, particularly in the subfield of AAC [17], as evidenced by many AAC companies attempting to include AI software into their products. However, empirical research on the effective integration of AI into AAC tools has not been well studied [18]. This study aims to address some of these gaps in research. 

Usability testing is a critical element in the product development cycle of apps in mobile health (mHealth) and education [19]. There are a host of methods available for usability testing, including questionnaires, think aloud walkthrough, task completion, interviews, focus groups, heuristic testing, and automated methods. A recent scoping review [20] revealed that most usability studies in eHealth use a combination of at least two of these methods, and the overall order in terms of frequency of use was as follows: questionnaires (*n* = 105), task completion (*n* = 57), ‘Think-Aloud’ (*n* = 45), interviews (*n* = 37), heuristic testing (*n* = 18), and focus groups (*n* = 13).

Using a combination of quantitative and thematic analyses methods, the purpose of this study was to (a) determine which of two AI algorithms (NLG-AAC and GPT-3.5) results in more relevant vocabulary with the *QuickPic AAC* application; and to (b) evaluate the perceived usability of *QuickPic AAC* among practicing speech–language pathologists. 

## 2. Methods

### 2.1. Participants

Participants included eight speech–language pathologists (SLPs) ranging in age range from 25 to 64 years based in an outpatient pediatric hospital: four participants were between 25 and 34 years old, three participants were between 25 and 44 years old, and one participant was between 55 and 64 years old. In order to be included, participants had to meet the following criteria: (a) an active American Speech-Language-Hearing Association (ASHA) Certificate of Clinical Competence for Speech-Language Pathologists (CCC-SLP); (b) a minimum of one year of experience working with individuals who use AAC or individuals who might benefit from AAC; and (c) experience in having created at least one TSD. Participants were recruited based upon convenience sampling within an outpatient AAC center in the Northeast of the United States. Table 1 provides an overview of participant characteristics.

The Institutional Review Board considered this study as exempt because it is limited to research activities in which the disclosure of the human subjects’ responses outside the research did not reasonably place the subjects at risk of criminal or civil liability or was not damaging to the subjects’ financial standing, employability, educational advancement, or reputation. Participants provided verbal consent. 

### 2.2. Materials 

Materials included (a) a tablet (i.e., iPad Pro) and the *QuickPic AAC* iOS application; (b) *QuickPic AAC* Reference Guide (see Appendix A); (c) the Demographic and AAC Experience Questionnaire (see Appendix B); (d) a vignette; (e) photographs; and (f) two usability questionnaires. 

Tablet and *QuickPic AAC*. The *QuickPic AAC* app ran on an iPad Pro. The *QuickPic AAC* app evolved from the development of an earlier prototype described in Fontana de Vargas et al. [21]. To generate vocabulary automatically, *QuickPic AAC* employs two different approaches for generating vocabulary. The first approach, proposed by Fontana de Vargas and Moffat [21] (which has now been coined NLG-AAC), uses the Visual Storytelling Dataset (VIST) [22] as the main source of vocabulary. VIST is composed of 65,394 photos of personal events, grouped in 16,168 stories. Each photo is annotated with captions and narrative phrases that are part of a story, created by Amazon Mechanical Turk workers. The NLG-AAC method works by first identifying the photographs in VIST that are most similar to the input photograph. This is accomplished by calculating the sentence similarity between the input photo caption, generated using the computer vision technique from Fang et al. [23], and all VIST photos captions. Then, the method retrieves all stories associated with those photographs and finds the most relevant words to present in *QuickPic AAC* by applying the Affinity Propagation clustering algorithm [24], and finally, gathering the top 50 most frequent words in the identified clusters.

The second approach, named GPT-AAC, takes advantage of recent advancements in natural language processing (NLP), a subfield of AI. More specifically, the method prompts the large language model (LLM) GPT-3.5 to produce the desired set of words related to the input photo caption (which is created using the method from Fang et al. [23], as in NLG-AAC). The prompt used by the method is shown below:

“You are a Speech Language Pathologist specialized in Augmentative and Alternative Communication.”

“Your task is to provide vocabulary related to a situation to help a person with communication disability to formulate messages about the situation. This vocabulary must contain words that people would often use to talk about that situation, either to describe it as well as to tell a story about it.” 

“The vocabulary must contain 20 verbs, 20 descriptors (adjectives and adverbs not terminating with LY), 20 objects, and 15 prepositions.” 

“All words must be in the first person singular, infinitive form without ‘to’.”

In both NLG-AAC and GPT-3.5, the underlying codes, as noted above, remain the same for generating vocabulary output for the TSDs. When participants customize the settings related to vocabulary (i.e., number of icons shown per part of speech), the vocabulary displayed to the participants is adjusted according to the specific customizations assigned in the settings. This allows the display to be customized for each user based upon the number of items per part of speech (e.g., subjects, verbs, etc.) while the underlying prompt continues to be consistent and guide the vocabulary generation process. Readers interested in the app design process through the lens of human–computer interaction research may reference the paper by Fontana de Vargas et al. [18]. 

The *QuickPic AAC* Reference Guide. The *QuickPic AAC* Reference Guide is a set of instructions that is available within the app (see Appendix A). For the purposes of this paper, the following terminology is adopted to describe aspects of *QuickPic AAC* communication displays (see Figure 1): (a) topic-specific display: thematic or topic-specific vocabulary that is arranged in grammatical categories (subject, verb, object, etc.); (b) static scene cue: a photograph of a single activity and/or concept [4,25]; (c) mixed display: a display containing a scene cue combined with a topic-specific display [4].

Demographic and AAC Experience Questionnaire. The Demographic and AAC Experience Questionnaire (Appendix B) elicited key demographic data (e.g., years as a practicing SLP) and previous experience with AAC, including their perspectives on TSDs. 

Vignette. The vignette was a prewritten case study that informed participants of the context in which they would be creating the TSDs. This was provided to all participants to read prior to creation of a TSD with *QuickPic AAC*: 

You are a speech–language pathologist in an outpatient pediatric setting and have a 7; 2 year old male patient with a primary diagnosis of autism spectrum disorder-level 3. Medical history includes no functional concerns regarding vision, hearing, or motor status. Receptive language skills include strong comprehension of noun-based vocabulary and ability to follow single-step directions within familiar contexts. Expressive language skills include scripted phrases (e.g., I want __), and single word approximations to label. Aided communication strategies include a grid-based communication application used primarily for requesting, labeling, and protesting. A goal of speech therapy is commenting/describing using 3 word utterances. A highly preferred activity/topic of conversation are cars/trains. Based upon this case study, create a QuickPic AAC display revolved around cars/trains using the ‘search [the web]’ function.

Photographs. As noted in the instructions within the vignette, participants were asked to choose one photograph for both conditions using the “Search the Web” feature of *QuickPic AAC*. One participant each chose a photo of a sports car on the road, and a photo of two boys playing trains together, respectively. Two participants each chose a photo of two boys playing race cars together, a photo of a boy playing with a wooden train set on the floor, and a photo of a boy playing with cars and trucks on a hardwood floor, respectively. Some participants chose identical photos from the web searches, likely because these images appeared first as initial search results.

Two Usability Questionnaires. The Mobile Health (mHealth) App Usability Questionnaire (MAUQ) [26] and a questionnaire adapted from Fontana de Vargas et al. [21] were administered. The MAUQ [26] (see Appendix C) was used to assess the usability of *QuickPic AAC* with its two approaches. The MAUQ has adequate psychometric characteristics and includes a 7-point Likert scale containing 16 items about interaction, vocabulary, and usage factors. The MAUQ was adapted minimally to meet the specific needs of our user study. Specifically, one question was eliminated (i.e., “I could use the app even when the Internet connection was poor or not available”) as the *QuickPic AAC* application requires internet connectivity. Additionally, one question was modified from “This mHealth app provides an acceptable way to deliver healthcare services, such as accessing educational materials, tracking my own activities, and performing self-assessment” to “This app provides an efficient way to create visual supports, such as educational, speech-language therapy, and language learning materials”.

The second questionnaire used in this study was adapted from Fontana de Vargas et al. [21] (see Appendix D). This questionnaire illustrates how the participants perceived the quality of three different areas in the application including: interaction, vocabulary quality, and overall usage. Modifications were made to serve the specific purposes of this study’s objectives. First, terminology was adapted across the entire survey from third person (e.g., “Users could easily select a desired vocabulary item within a page”) to first person language (e.g., “I could easily select a desired vocabulary item within a page”). Within the interaction subsection, two items related to creation of previous communication boards were eliminated as this did not pertain to the objectives of this research (i.e., Users tended to access/use vocabulary from previously created pages, Users tended to access/use vocabulary from newly created pages). Within the “Vocabulary” subsection, one question was modified from “The generated vocabulary included words users did not want to use” to “the vocabulary generated included words I would not have thought of that are relevant”. In addition, three items were added. Specifically, the items added were “The order the vocabulary was presented was adequate”, “The vocabulary generated included words I would target during educational and/or speech therapy sessions”, and “Overall the vocabulary generated is effective in helping me achieve targeted goals for my use”. Lastly, within the “Usage” subsection, one item was modified. Specifically, “Users were more communicative using the application than they usually are using other AAC tools” was modified to “I created topic-specific displays using this application more efficiently than with other AAC tools”. In addition to the two questionnaires, five open-ended questions related to overall experience and vocabulary generation across the two conditions were utilized. 

### 2.3. Design and Measures

A descriptive usability study was completed to evaluate the feasibility of using AI to generate relevant vocabulary for TSDs. This prospective design is consistent with a case series [27] in that the SLPs were exposed to *QuickPic AAC* with the two AI approaches following the reading of the vignette and the outcomes were monitored with observations and via questionnaire. Two dependent variables were measured within this study: (a) specificity of the vocabulary generated across two AI conditions (i.e., the natural language generation (NLG) approach based on de Vargas and Moffatt [21]; and the GPT-3.5 approach) and (b) user satisfaction. The specificity of the vocabulary generated was measured in terms of percentages as follows: (a) vocabulary/icons *kept* by the participant for the final TSP relative to vocabulary/icons originally produced by AI; (b) vocabulary kept for the final TSP but with icons *altered* by the participant out of the total # of vocabulary kept (alteration may involve the participant choosing a different icon to represent the vocabulary identified or moving the existing icon to a different column in the display); and (c) vocabulary/icons *deleted* by the participant from the final TSP relative to vocabulary/icons originally produced by AI (these two measures are inversely related).

Overall user satisfaction with each condition served as the second dependent variable, measured by two questionnaires as described previously. 

### 2.4. Procedures

#### 2.4.1. Demographic and AAC Experience Questionnaire

Upon enrollment in this study, participants completed a questionnaire regarding pertinent demographic information and previous experience with AAC (Appendix B). In addition, two brief questions were asked regarding their perspectives towards the benefits of TSDs and the challenges behind the creation of TSDs.

#### 2.4.2. Tutorial 

Participants engaged in a two-part tutorial process. Participants initially were provided a printed-out *QuickPic AAC* Reference Guide and were asked to read through the reference guide independently to familiarize themselves with the functions of *QuickPic AAC*. Subsequently, each participant individually took part in a live tutorial session led by the examiner, during which each feature in the reference guide was demonstrated including: creating a new board, editing a board (i.e., add, delete icons), editing an individual button, changing an individual button’s background color, locating a saved board, customizing “My Album”, and tips and tricks to create boards. Participants were able to use any of the features listed in the *QuickPic AAC* Reference Guide to customize their TSD, including adding icons, modifying/editing an existing icon, deleting icons, rearranging icons, etc.

#### 2.4.3. Experimental Task 

Following the tutorial phase, each participant received instructions to generate two TSDs with *QuickPic AAC* utilizing two separate approaches. Participants were aware the purpose of this study was to determine which approach generated more appropriate vocabulary. The two approaches encompassed the NLG method and the GPT-3.5 model. Participants remained blind to both conditions, and the sequence of conditions were randomized amongst participants to mitigate potential order-related effects. The creation of TSDs under both conditions for all participants were screen recorded using the built in screen recording feature within the iPad. To initiate a recording, the control center was enabled via the examiner by swiping down from the upper-right corner of the screen and selecting the “Screen Recording” icon. The standard, built in iOS setting of a three-second countdown signaled the start of the recording to the participants. The examiner stopped the recording when participants indicated they were completed with each TSD. For anonymity purposes, the recordings did not include sound and only captured the visual content on the screen. This allowed for data analysis to identify the vocabulary selections deemed relevant by participants across both conditions. Participants were provided with explicit instructions for using the app based on the *QuickPic AAC* Reference Guide (see Appendix A) and previously described vignette to create two identical outputs under the two conditions. Additionally, participants were instructed to determine the settings of the app that best suited the child depicted in the vignette, including the number of items populated within each part of speech (e.g., subjects, verbs, prepositions, descriptors, and objects), number of columns available of each part of speech, message bar size, and size of the input photo.

#### 2.4.4. Usability Questionnaires

Following their participation in the creation of two mixed displays, participants individually completed a modified version of the MAUQ and a post-questionnaire. These questionnaires were completed independently either directly after the *QuickPic AAC* experience or submitted to the experimenter no later than 24 h subsequent to their usage of the *QuickPic AAC* application. The questionnaires needed to be completed within this timeframe to ensure that participants’ experiences and impressions of the task remain recent in order to gain accurate and reliable feedback. This alleviates recall bias, which can occur if participants forget details, as well as helps prevent participants from discussing their experience with others, mitigating social desirability bias. This standardized response window that was maintained across all participants enhanced the comparability of the data. The post-questionnaires allowed for participants to provide their experiences of the two conditions facilitated by the NLG-AAC approach and the GPT-3.5 approach.

### 2.5. Data Analysis

Data on the perceived benefits and barriers to creating TSDs (AAC Experience Questionnaire) were analyzed descriptively (the small sample size precluded statistical analysis) by calculating the number of participants who were in support of statements on benefits and barriers, respectively. 

Relevant vocabulary was analyzed using simple descriptive summary statistics for each of the approaches (NLG-AAC and GPT-3.5) in terms of specificity. This includes the range, mean, and standard deviations (SD) of the ratios (i.e., percentages) of the vocabulary kept, the vocabulary deleted, and the icons that were modified. As sample size was small, the data were analyzed using the Friedman nonparametric test for several related samples [28]. This test analyzes data for significant differences among the mean ranks for the dependent variables (i.e., vocabulary kept, vocabulary deleted, vocabulary/icons modified). Significant differences were analyzed using the Wilcoxon Signed Rank test [29]. The Bonferroni correction was applied in order to reduce the type 1 error.

Data on overall usability were also analyzed using simple descriptive summary statistics for both surveys (MAUQ and post-questionnaire) for each of the conditions (NLG-AAC and GPT-3.5), including the range, mean, and standard deviations (SD) of the scores in both surveys. Further analysis was conducted within the post-questionnaire. Item analysis was achieved by calculating means across all eight participants per item. Sub-group analysis was also achieved by calculating means across items within the three subgroups. Finally, thematic analysis on the open-ended questions was conducted to reveal overall usability. 

## 3. Results

### 3.1. Perspectives on Benefits of and Barriers in Creating TSDs

Users’ perspectives on the perceived benefits of and barriers to creating TSDs in general (i.e., without *QuickPic AAC*) were revealed through an analysis of the AAC Experience Questionnaire. Participants responded to perceived benefits of TSDs (Table 2) and barriers in creating TSDs without *QuickPic AAC* (Table 3). At a group level, 8/8 (100%) participants agreed with the following benefits of TSDs: (a) facilitates expansion of utterance length, (b) helps with addressing communication goals in sessions, (c) helps with modeling of vocabulary, and (d) increases my client’s ability to communicate about a specific topic. Additionally, 6/8 (75%) participants agreed that TSDs increased the fluidity of communicating about a specific topic at hand. In terms of barriers to creating TSDs without *QuickPic AAC*, 8/8 (100%) of participants reported the time it takes to create TSDs was a barrier to including them in sessions. There was more discrepancy in relation to other perceived barriers, including that (a) 3/8 (37.5%) participants found it challenging to create visually appealing TSDs and they were unsure of the organization, framework, and guidelines for creating TSDs to include in sessions; (b) 2/8 (25%) participants reported that it was challenging to identify vocabulary and language to target using TSDs, while 1/8 (12.5%) participants reported that they did not have the resources (i.e., apps, software) to create TSDs. 

### 3.2. Group-Level Descriptive Results 

Vocabulary/Icons Kept. Participants were asked to read the vignette and create TSDs under both conditions using the *QuickPic AAC* app. Vocabulary/icons kept by the participants ranged from 6.67% to 64.52% (M = 38.55%; SD = 20.45%) for NLG-AAC and from 33.33% to 100% (M = 58.04%; SD = 21.89%) for GPT-3.5. Across 6/8 or 75% of participants, a greater percentage of vocabulary/icons was kept in the GPT-3.5 condition (Figure 2). 

Vocabulary Kept, but with Icons Altered. Some vocabulary was kept by participants but either they chose to alter the icons representing the vocabulary item or they chose to place the icon into a different column of the Fitzgerald Key layout of *QuickPic AAC*. Icons altered by the participants ranged from 0% to 6.25% (M = 3.38%; SD = 2.97%) for NLG-AAC and from 0% to 31.25% (M = 5.01%; SD = 10.82%) for GPT-3.5. Thus, slightly more icons were kept but altered with GPT-3.5. 

Vocabulary/Icons Deleted. Vocabulary/icons deleted by the participants ranged from 35.48% to 86.67% (M = 58.06%; SD = 19.23%) for NLG-AAC and from 0% to 66.67% (M = 36.94%; SD = 23.34%) for GPT-3.5. Thus, significantly more vocabulary was deleted with NLG-AAC relative to GPT-3.5.

### 3.3. Group-Level Inferential Results

A Friedman test was conducted to determine if there were statistical differences across conditions (i.e., NLG-AAC, GPT-3.5) among the mean ranks of the vocabulary kept, the vocabulary deleted, and the vocabulary modified. A statistically significant difference was found; X^2^ (5, *n* = 8) = 26.113, *p* < 0.001. This indicates there were differences among the six mean ranks. Three orthogonal contrasts were performed with Wilcoxon tests. For vocabulary kept, the contrasts between NLG-AAC (M rank = 3.88) and GPT-3.5 (M rank = 4.75) were significant (*p* < 0.05). For vocabulary deleted, the contrasts between NLG-AAC (M rank = 5.13) and GPT-3.5 (M rank = 3.88) were significant (*p* < 0.05). For vocabulary modified, no significant difference was observed between NLG-AAC (M rank = 1.94) and GPT-3.5 (M rank = 1.44) (*p* < 0.05). 

### 3.4. Individual Participant Results

In addition to examining group-level data, it is pertinent to examine participant-level data. Figure 3 displays finalized TSDs per participant created with each condition and the individual vocabulary/icons kept (circled in red), kept but modified (circled in yellow), or deleted (circled in blue) (Appendix E).

Overall Usability. All eight participants completed two post-questionnaires comparing their experiences between NLG-AAC and GPT-3.5 conditions related to overall experience and satisfaction. Results from the MAUQ are depicted in Figure 3. On a group level, usability scores ranged from 2.41 to 7.00 (M = 4.77; SD = 1.33) and from 4.12 to 7.00 (M = 5.47; SD = 0.86) for the NLG-AAC condition and the GPT 3.5 condition, respectively.

The second post-questionnaire participants completed was adapted from Fontana de Vargas et al. [23]. Results from this post-questionnaire are shared in Figure 4. Overall usability scores for the NLG-AAC condition ranged from 3.69 to 5.38 (mean = 4.80, SD = 0.64) while the GPT-3.5 condition ranged from 4.12 to 6.75 (mean = 5.82, SD = 0.65). These scores demonstrate overall higher usability scores for the GPT-3.5 approach, reinforcing those obtained from the MAUQ scores, demonstrating overall higher usability scores on GPT-3.5 condition.

To give a more detailed perspective, the post-questionnaire results were also analyzed at the item level and sub-group level (i.e., interaction, vocabulary generation, and overall usage). Figure 5 provides these results in detail. Item analysis was obtained by calculating averages across all eight participants per item. Sub-group analysis was also obtained by calculating averages across items within the three subgroups: interaction, vocabulary generation, and overall usage. Most prominently, the vocabulary generation sub-group demonstrated the most noticeable difference between NLG-AAC and GPT-3.5 conditions, with an overall greater score in the GPT-3.5 condition.

Lastly, participants were asked open-ended questions about their experience using *QuickPic AAC*. Results from the open-ended questions on overall experience are presented in Appendix F, while use case scenarios are provided in Table 4 from all of the participants. All responses are reported verbatim from the participants, unless indicated otherwise through the inclusion of brackets. Participants were randomly assigned and were unaware of each condition, and reported on their experience between Experience A and Experience B. The use of brackets was employed to clarify the condition (i.e., NLG and GPT-3.5) being referenced by each participant. 

Responses across participants reveal a general unanimous consensus on the feasibility and usability of *QuickPic AAC* in creating TSDs. An overall theme across participant reports demonstrated that the app offered a quick and easy way to create TSDs. Notably, two participants even commented that their experience with *QuickPic AAC* surpassed alternative AAC apps (i.e., Boardmaker, TouchChat HD-AAC). Users noted that it was beneficial that *QuickPic AAC* provided a starting point to create TSDs, facilitating the rate at which TSDs could be created. Lastly, users commented on *QuickPic AAC*’s intuitive interface, emphasizing its ease of use and the ease of the editing process. These responses overall demonstrate *QuickPic AAC*’s ability to streamline TSDs.

## 4. Discussion

As artificial intelligence (AI) makes significant headway in various arenas, the field of speech–language pathology is at the precipice of experiencing a transformative shift towards automation. This study aimed to introduce *QuickPic AAC*, an AI-driven application designed to generate topic-specific displays (TSDs) just-in-time from photographs. Specifically, the purpose of this study was to (a) determine which of two AI algorithms (NLG-AAC and GPT-3.5) results in more relevant vocabulary with the *QuickPic AAC* application; and to (b) evaluate the perceived usability of *QuickPic AAC* among practicing speech–language pathologists. The data provide statistically significant evidence that GPT-3.5 consistently generates more relevant vocabulary in that it consistently results in more vocabulary kept for final TSDs and in less vocabulary/icons deleted. It is noteworthy to indicate that the more vocabulary that is kept, the less editing is needed, and therefore less time is needed to create personalized TSDs. In general, SLPs expressed an overall high satisfaction in using *QuickPic AAC*. *QuickPic AAC*’s ability to swiftly create user-friendly TSDs may pave the way for other AI-driven tools to enhance language intervention strategies.

A primary focus of this study was on the quality of appropriate vocabulary generated through a specific controlled use case scenario (i.e., vignette). Overall, our findings provided insight that different AI algorithms provide varied vocabulary based on the same stimuli (i.e., a photograph). Also, in general, the GPT-3.5 algorithm provided more relevant vocabulary based upon SLPs’ judgments. A noteworthy discussion point that can be highlighted from the results is the large SD in the percentage of relevant vocabulary kept for both conditions, suggesting a wide variation in the number of vocabulary items that participants deemed relevant to keep. Some participants retained a relatively low percentage of icons (i.e., NLG-AAC: 6.67%, GPT-3.5: 33.33%), while others kept a significantly higher percentage (i.e., NLG-AAC: 64.52%, GPT-3.5: 100%). From a clinical standpoint, this poses an interesting finding as it indicates perceived relevance or important of vocabulary may not be consistent amongst SLPs.

While the statistical analysis offers valuable insights, there are additional interesting qualitative observations that may be further analyzed and discussed. For example, the symbol “baby” (to represent the vocabulary “young”) and “old man” (to represent the vocabulary “old”) frequently appeared in the vocabulary generated by the NLG-AAC algorithm, but was consistently deleted by all participants. All participants in this study deleted the icons “old” and “young”, indicating the vocabulary was not appropriate to target for this particular child described in the vignette. Additionally, from an app programming perspective, the NLG-AAC algorithm tended to generate “old” and “young” when the algorithm identified that there was a child present in the photo. In contrast, the GPT-AAC algorithm did not generate these words, which aligns with the greater satisfaction reported for GPT-AAC from a clinical standpoint. This qualitative observation further highlights our findings that the GPT-AAC algorithm generates more contextually appropriate vocabulary as deemed by SLPs, suggesting it may be more effective in clinical applications.

Another interesting observation involves the vocabulary generation from the algorithms when comparing photographs with and without a human figure. Specifically, Participant #1 selected a photograph of a car that does not feature any people, while all of the other participants selected photographs of vehicles of some sort with at least one person. Seemingly, this stands out as having the lowest number of deleted icons across all NLG-AAC trials. This trend is also consistent across GPT-3.5 trials as well, with the exception of Participant #8, who did not delete any items when using the GPT algorithm. This suggests that the absence of humans in the photographs selected may influence the relevance of the generated vocabulary, leading to fewer deletions and potentially indicating more appropriate vocabulary generation. Future studies may focus on the selection of photograph stimulus with and without humans and comparing relevance of vocabulary content. 

Further, it is important to consider the consistency of algorithm-generated vocabulary across different participants who selected the same images. A total of five different photographs were selected by the eight participants, serving as input stimuli. That means three photographs were selected by more than one participant allowing for a comparison of algorithm consistency (NLG-AAC, GPT-3.5). Specifically, participants #4 and #8 both selected a photo depicting a boy playing with trains on a track, participants #3 and #5 both selected a photo of two boys playing with race cars, and participants #6 and #7 both selected a photo of a boy playing with cars on a wooden floor. There was variability in the TSD arrangement (i.e., grid size, number of columns assigned per part of speech, and number of icons generated per part of speech) due to participants being instructed to adjust the settings to best suit the child depicted in the vignette. However, there was no variability in the vocabulary being generated by both algorithms when the same photograph and settings were selected. For example, participants #4 and #8 both selected the photograph of “the boy in the green shirt playing cars on the wooden floor”. In the NLG-AAC condition, Participant #4’s settings included up to four icons and one column per part of speech, while Participant #8’s settings included up to eight icons and two columns per part of speech. Despite these differences affecting the aesthetics of the TSD, all of the icons generated in Participant #4’s TSD were also generated in Participant #8’s TSD and in the same order. This was observed consistently across both NLG-AAC and GPT-3.5 conditions, in all three instances. This is further confirmed as Participant #3 and Participant #4 both selected the photo of “two boys playing race cars”. In the NLG-AAC condition, both participants’ settings were the same (i.e., up to four icons for each part of speech) and the vocabulary generated across both participants was consistent and in the same order.

Because the photographs were the same within each participant (for both conditions) we successfully controlled for threats to internal validity due to item difficulty for within-participant comparisons between the two conditions (NLG-AAC and GPT-3.5).

Another primary focus of our study was the overall satisfaction and usability of *QuickPic AAC* amongst SLP professionals. As discussed previously, the personalized creation of TSDs has a myriad of benefits reported by speech–language pathologists. These advantages include the expansion of utterance length, aiding clinicians in targeting specific communication goals and objectives during sessions, facilitating effective vocabulary modeling, supporting aided language stimulation, and increasing ability to communicate about a specific topic or activity. While the advantages are apparent, certain barriers were identified in the integration of TSDs, with the largest barrier of all being the time constraints as a primary obstacle in incorporating TSDs into SLPs’ sessions. Our overall findings demonstrate SLPs were satisfied in using an AI-generated app to create TSDs as it was a quick and efficient way to personalize communication materials for their clients. 

## 5. Limitations and Future Directions

While our preliminary findings are promising in demonstrating the use of AI in speech–language pathology to create TSDs, there are several limitations that need to be recognized. First and foremost, one limitation pertains to the use of different photographs across participants. With the exception of the participant pairs who received the same photos (as described above) they were not kept consistent across all eight participants. Thus, the nature of the photographs may have introduced an extraneous variable that influenced the outcomes. Future research should keep the photographs constant across participants or match the nature of the photos across participants. Relatedly, it is yet unknown whether the nature of the scenes displayed in the photographs afford better or worse AI-powered generation of vocabulary. In the current study, participants used *QuickPic AAC* with only one input photograph across two algorithms. Future research should strive to have participants use multiple input photographs to enhance external validity.

Similarly, the current design allows participants to engage with *QuickPic AAC* utilizing different settings (i.e., selecting the number of icons per part of speech), which may introduce variability in their decision-making processes. Future studies should strive to have standardized settings to parse out usability of the application itself versus appropriateness of vocabulary generated by the AI algorithms.

An additional limitation pertains to the small sample size and single-site recruitment. The eight participants were recruited at the same institution to allow for consistent implementation and evaluation of the application within a controlled environment. As an exploratory study, our goal was to gain insights into the potential benefits of two AI algorithms to create TSDs. As such, a smaller sample size was appropriate to meet these objectives and to identify whether further studies were warranted. This leads us to further discuss the heterogeneous characteristics of the participants in this study (e.g., chronological age, years practicing as an SLP, frequency working with individuals who use AAC, etc.). For instance, frequency of working with AAC users may play an influential role in their familiarity with various AAC tools and applications and their comfort level in interacting with technologies. Years of experience may impact their understanding of perspectives and strategies of using AAC tools. Having a broad range in characteristics likely reflects the population characteristics of those clinicians working in hospital settings and therefore is a net positive. However, the small sample size does negatively impact generalizability and requires future research to expand the external validity of findings. 

This leads us to further discuss the heterogeneous characteristics of the participants in this study (e.g., chronological age, years practicing as an SLP, frequency working with individuals who use AAC). Importantly, *QuickPic AAC* is meant to be an additive AAC tool that is used in conjunction with an individual’s primary AAC system to enhance communication related to specific topics of interest. Examples of use could be sharing information about weekend news, describing an activity that occurred at school, discussing a highly preferred area of interest, etc. By bridging the use of *QuickPic AAC* with an individual’s primary communication tool, this may provide an avenue for enriched personalized instruction and opportunity to capitalize on teachable moments. 

There are some directions in terms of development as well. It is essential to acknowledge a notable restriction of *QuickPic AAC*, specifically its reliance on internet connectivity for functionality, as it uses GPT-3.5. This limitation restricts its usage to environments that are equipped with internet access only. This issue highlights the need for future improvement that can enhance its utilization across a broader range of settings. Given our findings that demonstrate GPT-3.5 provides superior relevant vocabulary than NLG-AAC, this study needs to be expanded to include comparison of other AI data sets. Exploring different AI models and their effectiveness in generating relevant vocabulary would likely provide a more comprehensive understanding of the capabilities and limitations of AI being used for vocabulary selection in AAC software and applications. Additionally, at this juncture, it would be of value to compare the performance of AI with the performance of humans (i.e., clinicians) in creating topic-specific displays. Furthermore, research should examine how *QuickPic AAC* can be implemented in practice settings involving minimally speaking individuals. Lastly, ethical considerations should be taken into account when integrating AI into AAC practices. Future studies should address issues such as privacy, biases, and security risks.

## 6. Conclusions

AI has considerable potential in allied health fields including speech–language pathology. In this study, an AI-driven application designed to generate topic-specific displays from photographs on the fly, *QuickPic AAC*, was evaluated in terms of the relevance of vocabulary generated using two different AI algorithms, and perceived usability. GPT-3.5 produced greater relevant vocabulary compared to NLG-AAC. Additionally, practicing SLPs rated *QuickPic AAC* high in terms of its usability in effortlessly creating topic-specific displays. By embracing AI-technologies such as *QuickPic AAC*, SLPs can leverage its capabilities to alleviate the time demands on creation of personalized materials and dedicate more attention to individualized care and treatment for improving communication skills in individuals.

## Figures and Tables

**Figure 1 ijerph-21-01150-f001:**
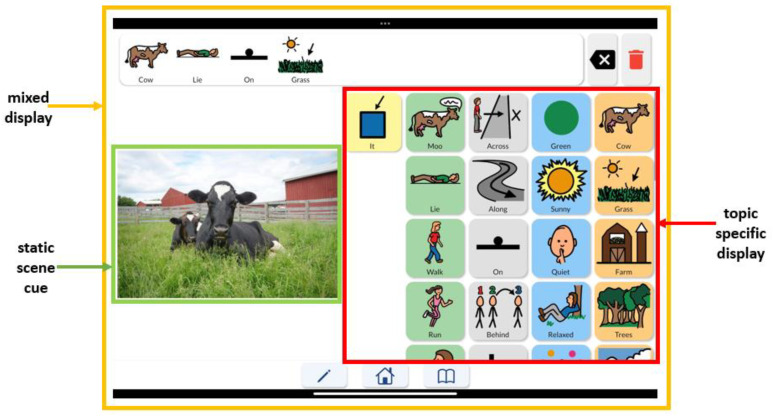
Guide describing components of a finalized communication display within QuickPic. © Copyright 1998–2022 Tobii Dynavox. All Rights Reserved.

**Figure 2 ijerph-21-01150-f002:**
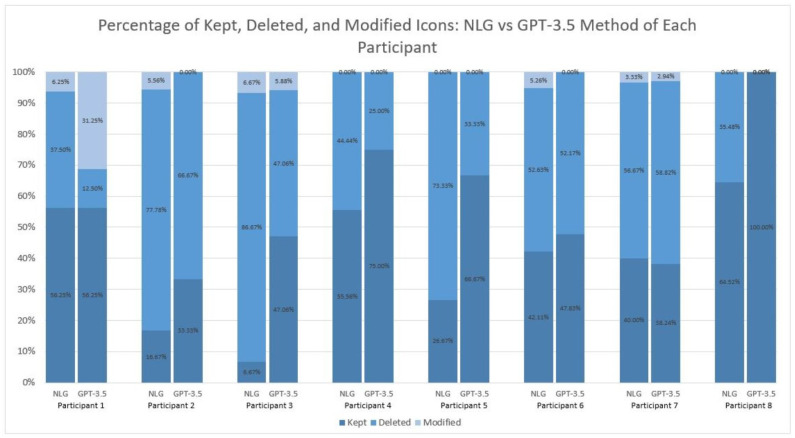
Percentage of Kept, Deleted, and Modified Icons of Each Participant: NLG-AAC vs. GPT-3.5 Method.

**Figure 3 ijerph-21-01150-f003:**
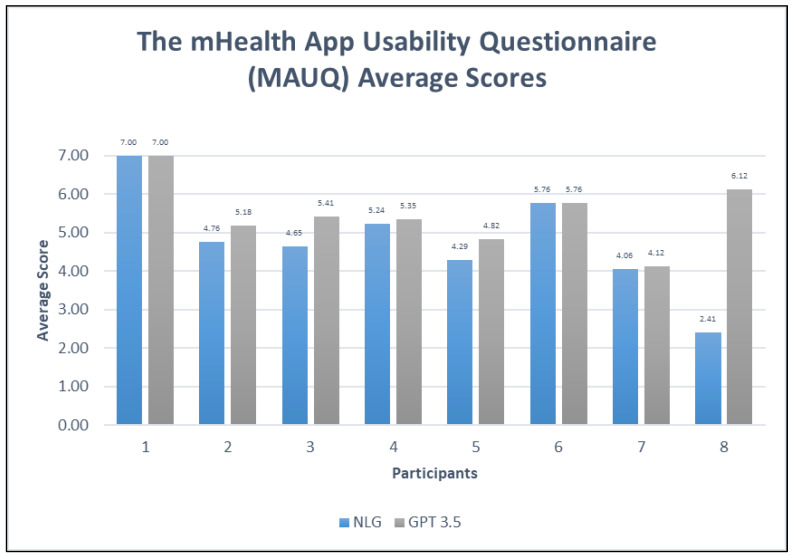
Participant overall average scores comparing the NLG-AAC to GPT-3.5 using the mHealth App Usability Questionnaire (MAUQ).

**Figure 4 ijerph-21-01150-f004:**
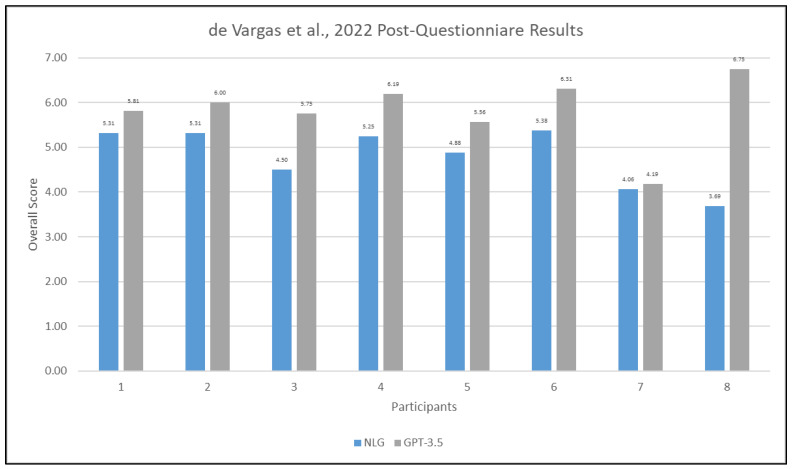
Bar graph demonstrating overall average scores from the de Vargas et al. [21] post-questionnaire survey results across all participants.

**Figure 5 ijerph-21-01150-f005:**
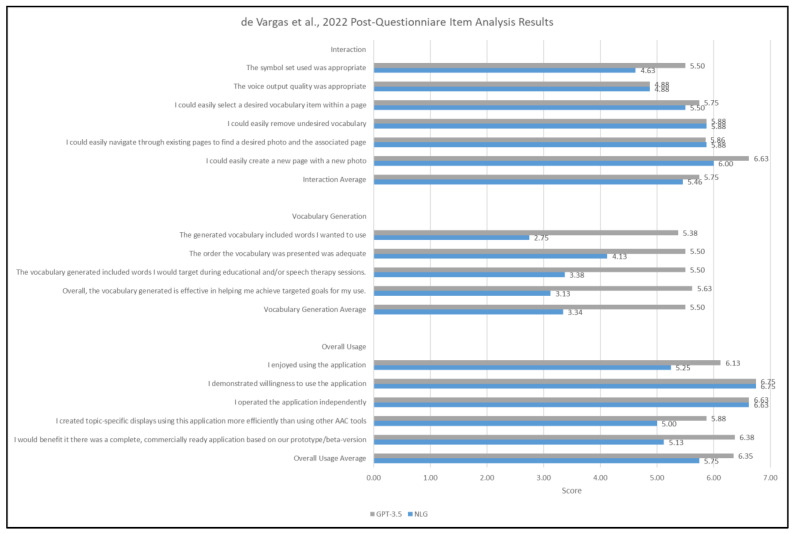
Bar graph demonstrating an item analysis and sub-group analysis from the post-questionnaire survey results across all participants from the de Vargas et al. [21].

**Table 1 ijerph-21-01150-t001:** Participant characteristics.

Participant	Race	Ethnicity	CA	Years Practicing as an SLP	Frequency of Working with Individuals Who Use AAC	Have You Created Topic Displays	Frequency of Creating Topic Displays	Average Length to Create Topic Displays
1	White	Not Hispanic or Latino	25–34	2	Weekly	Yes	Occasionally	31–40 min
2	More than one race	Not Hispanic or Latino	25–34	4	Daily	Yes	Weekly	11–20 min
3	White	Not Hispanic or Latino	35–44	17	Daily	Yes	Monthly	<10 min
4	White	Not Hispanic or Latino	35–44	12	Daily	Yes	Weekly	11–20 min
5	White	Not Hispanic or Latino	25–34	2	Weekly	Yes	Monthly	21–30 min
6	White	Not Hispanic or Latino	35–44	12	Daily	Yes	Daily	<10 min
7	White	Not Hispanic or Latino	25–34	6	Daily	Yes	Monthly	11–20 min
8	White	Not Hispanic or Latino	55–64	35	Monthly (varies)	Yes	Occasionally	51–60 min

Note. CA = chronological age; SLP = speech–language pathologist; AAC = augmentative and alternative communication.

**Table 2 ijerph-21-01150-t002:** Perceived benefits of topic-specific displays.

Participant	Increase my Client’s Ability to Communicate about a Specific Topic	Help Me Model Vocabulary	Help Me Address Communication Goals in My Sessions	Help Facilitate Expansion of Utterance Length	Help Increase the Fluidity of Communicating about a Specific Topic at Hand	Other
1	Yes	Yes	Yes	Yes	Yes	--
2	Yes	Yes	Yes	Yes	Yes	Help person who uses AAC to attend to the display and [create] word combinations; no dynamic display needed to navigate
3	Yes	Yes	Yes	Yes	Yes	--
4	Yes	Yes	Yes	Yes	Yes	--
5	Yes	Yes	Yes	Yes	No	--
6	Yes	Yes	Yes	Yes	Yes	Help direct families/teams re. modeling
7	Yes	Yes	Yes	Yes	Yes	--
8	Yes	Yes	Yes	Yes	No	--

**Table 3 ijerph-21-01150-t003:** Perceived barriers to creating topic-specific displays.

Participant	The Time It Takes to Create a Topic-Specific Display Is a Barrier to Including Them in My Sessions	It Is Challenging to Identify Vocabulary and Language to Target Using Topic-Specific Displays	I Do Not Have the Resources (i.e., Apps, Software) to Create Topic-Specific Displays	I Feel Unsure about the Organization, Framework, and Guidelines for Creating Topic-Specific Displays to Include Them in My Sessions	It Is Challenging to Create Visually Appealing Topic-Specific Displays	Other
1	Yes	Yes	Yes	Yes	Yes	--
2	Yes	No	No	No	Yes	--
3	Yes	No	No	No	No	It might prevent generalization of vocabulary across contexts
4	Yes	Yes	No	No	Yes	--
5	Yes	No	No	Yes	No	--
6	Yes	No	No	No	No	--
7	Yes	No	No	No	No	--
8	Yes	No	No	Yes	No	--

**Table 4 ijerph-21-01150-t004:** Open-ended questions related to use case scenarios and frequency of use across all participants.

Participant	If You Had Access to *QuickPic AAC*, Would You Incorporate It into Your Practice?	If Yes, How?	How Often Would You Use It?
1	Yes	NA	Weekly or monthly depending on my caseload
2	Yes	During therapy sessions to base my therapy on patient’s interests	Weekly
3	Yes	With communicators who need explicit support and phrase generation and have trouble navigating across pages	NA
4	Yes	Creating displays “on the fly” in therapy for common activities	Weekly
5	Yes	I would use it to create activity specific topic displays in a much more efficient manner. It would help me increase aided language modeling in sessions.	I regularly see patients who use AAC, so I would use it weekly.
6	Yes	For topic display users and families who are ready to start making their own.	In evaluations on a weekly/daily basis. As a recommendation, every other week.
7	Yes	Help families independently select vocabulary at home	Frequently
8	Absolutely	Creating topic display boards	My patient population is variable. I don’t always have patients who need topic display boards. I would use it anytime I needed to create a topic display board.

## Data Availability

Data are available upon request from the first author.

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
