# Peer review of "QuickPic AAC: An AI-Based Application to Enable Just-in-Time Generation of Topic-Specific Displays for Persons Who Are Minimally Speaking"

_ijerph, 2024, doi:10.3390/ijerph21091150_

Round 1
Reviewer 1 Report
Comments and Suggestions for Authors
Thank you for the opportunity to review this manuscript. I had an opportunity to learn about the QuickPic AAC App at a convention last year and it is good to see this article with evidence supporting its use. I appreciated the background information on the App and your methods to address usability. AI seems so frequently discussed in more abstract opportunities than actual products these days - I again appreciate your work. I do have some concerns with the current manuscript. I believe clarification could make for something stronger in understanding the underlying algorithms and in how the product can be used. INTRODUCTION
- Since your method involves decision-making by participants, I recommend you include and discuss the following reference or one like it in the literature review
Thistle, J. J., & Wilkinson, K. M. (2020). Speech-Language pathologists’ decisions when designing an aided AAC display for a compilation case study of a beginning communicator. Disability and Rehabilitation: Assistive Technology, 16(8), 871–879. https://doi.org/10.1080/17483107.2020.1745911 METHOD TASK I'm having difficulty understanding the experimental task. Page 7, lines 265-270 state, "Participants were provided with explicit instructions for using the app based on the previously described Vignette to create identical outputs under the two conditions. Additionally, participants were instructed to determine the settings of the app that best suited the child depicted in the vignette, including the number of items populated within each part of speech (e.g., subjects, verbs, prepositions, descriptors, objects), number of columns available of each part of speech, message bar size, and size of the input photo." Reading these sentences and the remainder of the paper, I was left with several questions that I recommend clarifying.
- Were the "explicit instructions for using the app" the same thing as the quick reference card? If not, I'd recommend sharing what the explicit instructions were.
- I'm also unclear on how the instructions/task relate to the "prompt used by the method" on p. 4 line 148-149 that specifies "The vocabulary must contain 20 verbs, 20 descriptors (adjectives and adverbs no terminating with LY), 20 objects, and 15 prepositions." Is that the underlying prompt that populates output using GPT 3.5? If so how does that interact with the setting that participants are able to control and make decisions about?
- For NLG-AAC on p. 4, line 135, what are the cutoff numbers in "gathering the most frequent words in the identified clusters"? How many are generated?
FIGURE 5 I'm not sure I fully understand Figure 5 and have multiple questions about it. It is possible the answers could negate some of the concerns, but I have raised them to the extent the impact my ability to understand the data set.
- There are items in the finalized version that are not shown in the original, but the manuscript states that participants did not have the ability to add items (p. 20 line 493). This happens across multiple participants. Just as one striking example, it looks like participant 3 deleted almost ever item in the original condition for NLG but there are 14 items in the final board. My guess is that the original side doesn't show all the items. At the very least that needs to be mentioned. It would be better to have a comprehensive list somewhere in the manuscript or in additional materials.
- On p. 19 line 451-452 the authors state, "There was no variability in the vocabulary being generated by both algorithms when the same photograph was selected." However when I look at Figure 5 and see the originals for participant 5 vs. participant 3, every item for participant 3 is in the output for participant 5, but there are also additional items for participant 5.
- Participant 4 in the GPT 3.5 final column has both up and down arrows. Neither of those items are in the original display. Only the down arrow appears to be in participant 8's GPT 3.5 original and final. There are also more items with participant 8 vs. participant 4.
- The challenges with Figure 5 lead me to a broader concern with the data. I believe you used percentages for the Friedman test. In descriptive statistics I can see how percentages help to compare the relative number of items kept or deleted. I am not certain of the statistical effect of having
VALIDITY
- I would be interested to know if there is any differences in order of presentation. For example, was there an effect for keeping or deleted items from the first to the second time regardless of AI condition?
- I was not clear if you controlled for multiple comparisons in your post-hoc testing. My understanding is that whether it is parametric or not, multiple comparisons still carry the same risk.
DISCUSSION
- There are two stated purposes to the study, the first relates to which AI algorithm results in the more relevant vocabulary with QuickPic AAC, and then the second is to evaluate the perceived usability. The experimental task has participants engage with the App 2 times but allows for different changes in settings to produce output. I wonder if you could discuss separating out the use of the APp from decision making on output? For example, could you have standardized the output for each condition and then had participants make judgements about what to keep or delete? I wonder if that could have helped to focus the decision-making aspect.
- It seems like there are potentially other interesting observations that are lost in the statistical analysis. For example...
- The baby came up frequently for NLG and for every participant it was deleted. The "old" person similarly came up frequently and was deleted by all but one participant.
- The photograph for participant 1 is the only one without any people. Comparatively it has the lowest number of deletions across all NLG trials and also all GPT trials with the exception of participant 8 who deleted no items for GPT
Author Response
Thank you for taking the time to review our manuscript. We appreciate your questions, clarifications, and feedback. Please see our responses in the attachment.

Reviewer 2 Report
Comments and Suggestions for Authors
1. Please state the research gap.
2. Several statements should cite the references. For example, “Many apps provide a range of tools that serve as a communication platform as well as a medium to provide language support.”
3. Please explain the reasons for choosing these three age ranges for the research participants.
4. These eight participants were all from one hospital. Please explain the reasons for recruiting only eight participants all from the same facility, in terms of generalizing research results nationally and internationally.
5. The case used in the study was a 7-year-old male patient (i.e., a child). Please explain the reason for the recruitment criteria not focusing on children who use AAC or children who might benefit from AAC.
6. In terms of participant characteristics, please explain the potential influence of these heterogeneous participant characteristics (i.e., Years Practicing as an SLP, Frequency of Working with Individuals Who Use AAC) on the results.
7. QuickPic AAC used photos from VIST. Please explain the potential considerations in choosing these photos from VIST, rather than other symbol sets/systems.
8. Please provide the reasons for choosing GPT-3.5 in this study rather than other approaches of AI.
9. Please explain why the provided prompt to GPT-3.5 is valid and appropriate.
10. The descriptions of Topic Specific Display in the QuickPic AAC Reference Guide should cite references.
11. Please provide the details of the “Demographic and AAC Experience Questionnaire” in the text.
12. Please explain the reasons for choosing the specificity of the vocabulary as one of the dependent variables.
13. The study had the participants aware of the study purpose. Please state if the awareness of the study purpose might affect the results.
14. Please state how to record the screen and who provided the instructions.
15. Please provide the reasons for choosing no later than 24 hours to fill the Usability Questionnaire.
16. There are only eight participants. Please consider using tables to represent the results, instead of figures (i.e., Figures 2 & 3).
17. Individual Participant Results represented in Figure 5 should be reported in the Appendix.
18. Table 2 should be reported in the Appendix.
Author Response
Thank you so much taking the time to review this manuscript. We appreciate your comments and feedback. Please find the detailed responses attached.

Round 2
Reviewer 1 Report
Comments and Suggestions for Authors
Thank you for the time taken to revise this manuscript and to highlight the changes. It made it easy to track how you responded to each suggestion. I have no further edits to suggest.
Reviewer 2 Report
Comments and Suggestions for Authors
none